# Investigating self-supervised representations for audio-visual deepfake detection

## Abstract

Self-supervised representations excel at many vision and speech tasks, but their potential for audio-visual deepfake detection remains underexplored. Unlike prior work that uses these features in isolation or buried within complex architectures, we systematically evaluate them across modalities (audio, video, multimodal) and domains (lip movements, generic visual content). We assess three key dimensions: detection effectiveness, interpretability of encoded information, and cross-modal complementarity. We find that most self-supervised features capture deepfake-relevant information, and that this information is complementary. Moreover, the models attend to semantically meaningful regions rather than spurious artifacts. Yet none generalize reliably across datasets. This generalization failure likely stems from dataset characteristics, not from the features themselves latching onto superficial patterns. These results expose both the promise and fundamental challenges of self-supervised representations for deepfake detection: while they learn meaningful patterns, achieving robust cross-domain performance remains elusive.

## 1 Introduction

Generative models now produce text, images, audio, video, all rivaling human creations. This progress brings a new challenge: detect whether content is authentic or machine-generated (deepfake). Reliable detection prevents obvious risks such disinformation and fraud, but also serves a simple need: users want to know what they can trust. We tackle the detection problem in the video domain, the internet's most consumed medium. Video is uniquely challenging because it spans both visual and audio modalities, demanding methods that can reason across them.

Many approaches are being continuously proposed for the task of audio-visual deepfake detection: these range from powerful discriminative classifiers (Koutlis & Papadopoulos, 2024a) to techniques that exploit inconsistencies between modalities (Feng et al., 2023). Yet, the backbone features underlying these models often determine their effectiveness. Recent work shows strong results of self-supervised learning in this context: image-based detectors benefit from CLIP (Ojha et al., 2023; Cozzolino et al., 2024), audio-based detectors from Wav2Vec2 (Wang & Yamagishi, 2022; Pascu et al., 2024), and audio-visual models from AV-HuBERT (Reiss et al., 2023; Liang et al., 2024). The self-supervised representations capture rich, modality-specific structure without requiring task-specific supervision, making them especially attractive for deepfake detection.

In this paper, we evaluate a wide range of self-supervised features (audio-only, image-only, multimodal) for the task of audio-visual deepfake detection. Our aim is to understand what these features capture and how they contribute to detection performance. We center our work around three research questions:

- **RQ1. Do self-supervised features encode useful information for deepfake detection?** Do they also generalize across domains and to the related task of anomaly detection?

- **RQ2. Where do self-supervised features look?** Does the model attend to the manipulated regions? Does it align with human annotations?

- **RQ3. How complementary are different features?** If multiple feature types succeed at detection, do they rely on similar cues or do they encode distinct information?

To answer these questions, we adapt linear probing (Alain & Bengio, 2017; Hupkes et al., 2018; Belinkov, 2022) to the video domain: we extract self-supervised representations, apply frame-level linear classifiers on top, and aggregate their predictions at the video level. By keeping the classifier minimal, we can directly measure the information already encoded in the feature representations. We find that a broad set of features (from visual encoders trained on generic images to speech-only models) encode deepfake information.

To assess whether this information is relevant, we conduct a multi-faceted evaluation. One way we probe the model's behavior is by extracting implicit localizations from the classifier. We use temporal and spatial explainability techniques to test whether these explanations align with regions that were actually manipulated. We further compare them to human annotations provided by Hondru et al. (2025). A second more conventional measure of effectiveness is out-of-domain generalization: we evaluate on three additional datasets, including in-the-wild data (Chandra et al., 2025). Finally, since even subtle distribution shifts between real and fake samples can lead to spurious correlations (Chai et al., 2020; Müller et al., 2022; Smeu et al., 2025), we also consider two tasks that rely on real data only: (i) density estimation via next-token prediction, and (ii) audio-video synchronization, which measures how well representations capture cross-modal alignment.

Our results show that many of the features align reasonably well with manipulation boundaries, and there is no obvious reliance on spurious features, such as background regions or silence in the audio. Still, out-of-domain performance remains inconsistent: even the strongest features struggle on the challenging in-the-wild dataset. The reasons for this generalization gap are likely tied to dataset characteristics, *e.g.* fundamentally different types of manipulations or uncertainty regarding the labels of the in-the-wild data.

## 2 RELATED WORK

**Self-supervised learning in audio and visual domains.** Self-supervised learning (SSL) learns powerful representations by solving pretext tasks on large-scale unlabeled data (Balestriero et al., 2023). These representations transfer effectively to many downstream tasks, such as classification or anomaly detection. In the **visual** domain (Uelwer et al., 2025), many approaches use contrastive learning to align images to their augmented version (MoCo, He et al. 2020; SimCLR, Chen et al. 2020) or images to their caption (ALIGN, Jia et al. 2021; CLIP, Radford et al. 2021). Alternatives include masked image modeling (MAE; He et al. 2022) or discriminative self-distillation (DINO, Oquab et al. 2023; Siméoni et al. 2025). For video, spatio-temporal information is used to build SSL representations (CVRL, Qian et al. 2021; VideoMAE, Tong et al. 2022). In the **audio** domain (Liu et al., 2022), the Wav2Vec family (Baevski et al., 2020; Babu et al., 2022) and HuBERT (Hsu et al., 2021) introduced predictive coding and masked prediction for speech signals, significantly improving ASR with limited labeled data. For **audio-visual** data, AV-HuBERT (Shi et al., 2022) extends HuBERT to learn joint speech and lip representations, while Auto-AVSR (Ma et al., 2023) integrates Conformer encoders for audio-visual speech recognition. These representations have been primarily used for speech-related tasks, but when trained on generic video they are also useful for localization (Akbari et al., 2021) or music classification (Wang et al., 2021).

**Self-supervised representations in deepfake detection.** Self-supervised representations have successfully transferred to deepfake detection. In the **visual** domain, Ojha et al. (2023) showed that a frozen CLIP backbone with only a linear layer yields state-of-the-art generalization performance. CLIP remains the most popular SSL encoder for image deepfake detection (Cozzolino et al., 2024; Srivatsan et al., 2023; Zhu et al., 2023; Khan & Dang-Nguyen, 2024; Koutlis & Papadopoulos, 2024b; Liu et al., 2024a; Reiss et al., 2023; Smeu et al., 2024), but others have explored other vision-language encoders (*e.g.*, BLIP2 by Reiss et al. 2023; Keita et al. 2025, InstructBLIP by Chang et al. 2023) or vision-only models (*e.g.*, DINO by Nguyen et al. 2024, MoCo by You et al. 2024). Similarly, the **audio** deepfake detection community has adopted large SSL representations, with most popular ones being Wav2Vec (Martín-Doñas & Álvarez, 2022; Wang & Yamagishi, 2022; Wang et al., 2022; Tak et al., 2022; Xie et al., 2023; Pianese et al., 2024; Pascu et al., 2024), followed by HuBERT (Kheir et al., 2025) and WavLM (Combei et al., 2024). For **audio-visual** deepfake detection, AV-HuBERT representations have shown strong performance not only in a fully supervised paradigm (Shahzad et al., 2023), but also in zero-shot (Reiss et al., 2023; Liang et al., 2024) or unsupervised (Smeu et al., 2025) settings.

**Audio-visual deepfake detection.** Audio–visual deepfake detection exploits inconsistencies between modalities or leverages joint representation learning for improved robustness. Early methods focused on cross-modal inconsistencies, such as phoneme–viseme mismatches (Agarwal et al., 2020) or emotional incongruence between speech and facial expressions (Mittal et al., 2020). Another line of work detects forgeries by modeling audio–visual synchrony, where misalignment between the two modalities signals manipulation (Feng et al., 2023; Smeu et al., 2025). More recent approaches emphasize joint learning and multimodal fusion: AVFF (Oorloff et al., 2024) aligns audio and video features with contrastive objectives, while SpeechForensics (Liang et al., 2024) use pretrained speech and lip representations. Other architectures combine modality-specific encoders via attention or transformers (Guo et al., 2023) or employ contrastive inter-modality differentiation (Koutlis & Papadopoulos, 2024a).

## 3 METHODOLOGY

Deepfake detection is a binary classification task that attempts to map an input video $\mathbf{x}$ to a binary label $y$ (1 if the video is fake and 0, otherwise). We build deepfake detection models that rely mostly on self-supervised features and learn a minimal amount of parameters on top of the features to perform the task of deepfake detection. The models have three steps: 1. extract locally temporal features (e.g., an embedding $\phi(\mathbf{x}_t)$ for each frame $t$); 2. apply a learnable linear classifier $\mathbf{w}$; 3. aggregate the predictions using a pooling function (such as log-sum-exp). Formally, the per-video score $s$ is defined as:

$$s(\mathbf{x}; \mathbf{w}) = \log \sum_t \exp \left\{ \mathbf{w}^\mathsf{T} \phi(\mathbf{x}_t) \right\} \tag{1}$$

Since the log-sum-exp approximates the max function, the model learns to predict that a video is fake if only a single region of the model is fake. The parameters of the linear layer are trained by minimizing the cross-entropy loss on video-level labels. See App. A.1 for implementation details.

Our approach is related to linear probing (Alain & Bengio, 2017; Hupkes et al., 2018; Belinkov, 2022): we keep the self-supervised backbones frozen and train only a simple model on top. This makes it possible to assess the quality of the representations in a comparable setting. While more complex models could be explored for downstream performance, prior work shows that simple linear models often suffice (Ojha et al., 2023; Pascu et al., 2024).

### 3.1 EXPLANATIONS

To understand how deepfake detection classifiers make decisions, we generate both temporal and spatial explanations.

**Temporal explanations.** Since the pooling function (log-sum-exp) is a simple transformation of its inputs, the final video-level prediction can be regarded as an aggregation of local frame-level predictions. We therefore compute per-frame scores $s_t = \mathbf{w}^\mathsf{T} \phi(\mathbf{x}_t)$ to measure which time segments contribute most to the final prediction.

**Spatial explanations.** Given that the per-frame classifier is linear, we can further decompose each frame-level decision into patch-level contributions. If the per-frame feature is computed by averaging patch features, we propagate the linear classifier down to the patch level (Zhou et al., 2016). If non-linear aggregation is used, we instead apply Grad-CAM (Selvaraju et al., 2017) to obtain patch-level relevance maps.

**Evaluation.** We compare the resulting temporal and spatial explanations against the annotated extent of local manipulations or human annotations. Since the classifiers are trained only with video-level supervision, this comparison also serves as a form of weakly-supervised localization.

### 3.2 PROXY TASKS

Instead of training a binary classifier, which risks latching onto spurious features, we explore two tasks that rely on real data only: next-token prediction and audio-video synchronization. These are proxy tasks because they do not address deepfake detection directly; rather, they model the distribution of real data, and assume that that deviations from this distribution indicate fakes. Both approaches have shown promising performance (Feng et al., 2023; Smeu et al., 2025).

| Model | Pretraining | | | Params. | Dim. |
|---|---|---|---|---|---|
| | Modality | Content | Dataset | | |
| *Audio features* | | | | | |
| Wav2Vec XLS-R 2B | audio | speech | MLS and others | 2159M | 1920 |
| Auto-AVSR (ASR) | audio | speech | LRS3 | 243M | 768 |
| AV-HuBERT (A) | multimodal | lips + speech | VoxCeleb + LRS3 | 310M | 1024 |
| *Visual features* | | | | | |
| Auto-AVSR (VSR) | visual (video) | lips | LRS3 | 250M | 768 |
| FSFM | visual (video) | faces | VGGFace2 | 86M | 768 |
| Video-MAE-large | visual (video) | generic | Kinetics-400 | 303M | 1024 |
| CLIP VIT-L/14 | visual (images) | generic | WebImageText | 303M | 768 |
| AV-HuBERT (V) | multimodal | lips + speech | VoxCeleb + LRS3 | 322M | 1024 |
| *Audio-visual features* | | | | | |
| Auto-AVSR | multimodal | lips + speech | LRS3 | 443M | 768 |
| AV-HuBERT | multimodal | lips + speech | VoxCeleb + LRS3 | 322M | 1024 |

Table 1: Overview of the self-supervised models studied in the paper. The "modality" column indicates the input streams used for the training objective. Note that this may differ from the modalities actually encoded.

**Next-token prediction** models the probability of the next frame's representation $\mathbf{x}_t$ given the previous frames $\mathbf{x}_1, \ldots, \mathbf{x}_{t-1}$. The assumption is that frames that cannot be predicted well are more likely to indicate manipulations. We use a decoder-only Transformer trained with the mean squared error on real videos. At test time, the model predicts each frame given its corresponding history, and we obtain a per-video fakeness score as the maximum frame-level mean squared error. The models' architecture has 4 layers each containing 4 heads, a feature dimension of 512 and a feedforward dimension of 1024. In order to match the input encoding dimension, projection layers are applied before and after the Transformer.

**Audio-video synchronization** models how well the audio and video frame-level representations match. The assumption is that mismatches between the two modalities indicate manipulations. We use an alignment network $\Phi$ (Smeu et al., 2025), where L2-normalized audio features $\mathbf{a}$ and visual features $\mathbf{v}$ are concatenated and passed through a four-layer MLP with Layer Normalization and ReLU activations. The network is trained to maximize the probability that an audio frame $\mathbf{a}_i$ aligns with its corresponding video frame $\mathbf{v}_i$, rather than with neighboring frames $N(i)$:

$$p(\mathbf{v}_i \mid \mathbf{a}_i) = \frac{\exp\left(\Phi(\mathbf{a}_i, \mathbf{v}_i)\right)}{\sum_{k \in N(i)} \exp\left(\Phi(\mathbf{a}_i, \mathbf{v}_k)\right)}. \tag{2}$$

At test time, the per-frame alignment scores $\Phi(\mathbf{a}_i, \mathbf{v}_i)$ are inverted to estimate fakeness, and then pooled across the video using a log-sum-exp operator to produce the final detection score.

## 4 MODELS

We study a wide range of self-supervised representation: from models trained on audio-visual data to models trained on visual-only or audio-only data. For the visual-only models, the content range from face models to generic images or video. Tab. 1 summarizes the models used in the paper.

### 4.1 VISION-ONLY ENCODERS

**CLIP** (Radford et al., 2021) is an image-text model trained on general images collected from the internet. For CLIP we use the ViT/L-14 model, and take the CLS token produced the image encoder. The CLS token performs a non-linear aggregation of patch features using attention and a MLP.

**FSFM** (Wang et al., 2024) is a foundation model trained on faces and reconstruction tasks. FSFM uses a ViT/B-14 model and performs average feature pooling. The model operates only on the face region, so it relies on a face detector.

**VideoMAE** (Tong et al., 2022) is a vision transformer model trained to reconstruct patches of generic video. The model is typically used for action recognition. It extracts features on spatio-temporal patches of size $2 \times 16 \times 16$ in a window of 16 frames. The window is moved with strides of 16. We discard the last window if its length is shorter than 16 frames.

### 4.2 AUDIO-ONLY ENCODERS

**Wav2Vec XLS-R 2B** (Baevski et al., 2020) is a speech model trained on 436k hours of multilingual data. The model employs a convolutional feature encoder followed by a Transformer, and learns to match context representations to the corresponding local quantized representations. We use the representations extracted from the last layer. Wav2Vec2 extracts features every 20 ms (50 Hz); to match a 25 FPS video, we concatenate each two consecutive feature vectors.

### 4.3 AUDIO-VISUAL ENCODERS

**AV-HuBERT** (Shi et al., 2022) is a Transformer that jointly models audio and visual features. Visual features are extracted with a 3D ResNet on lip regions, while audio features come from a feedforward network applied to log filterbanks. The model is trained with masked multimodal cluster prediction: masked audio-visual inputs used to predict automatically discovered and iteratively refined hidden units. We also extract modality-specific representations by masking one stream: audio-only features by masking the visual input (AV-HuBERT (A)), and visual-only features by masking the audio input (AV-HuBERT (V)).

**Auto-AVSR** (Ma et al., 2021; 2023) is an audio-visual speech recognition model. The model encodes the two modalities with a Conformer (Gulati et al., 2020): visual features are extracted with a 3D Resnet on lip regions, while audio features come from a 1D ResNet-18 followed by another Conformer. We also obtain modality-specific representations using independently-trained models: the visual-only branch trained exclusively on visual data (VSR), and the audio-only branch trained solely on audio data (ASR).

## 5 EXPERIMENTAL SETUP

### 5.1 DATASETS

We consider four audio-visual datasets varying in terms of generation methods, manipulation scope (full or local manipulation), source domain (controlled scientific settings or real-world media).

**FakeAVCeleb** (FAVC, Khalid et al. 2021) contains 500 real videos from VoxCeleb2 (Chung et al., 2018) and 19.5k fake videos, generated using face swapping methods (Faceswap, Korshunova et al. 2017; FSGAN, Nirkin et al. 2019), lip-syncing (Wav2Lip, Prajwal et al. 2020), and voice cloning (SV2TTS, Jia et al. 2018). Since the dataset does not come with a predefined split, in line with previous works we split the dataset in 70% (training and validation) and 30% testing samples.

**AV-Deepfake1M** (AV1M, Cai et al. 2024) comprises over one million videos of approximately 2k subjects. The videos are locally manipulated by modifying one or several words in the transcriptions with ChatGPT. Corresponding fake video segments are generated with TalkLip (Wang et al., 2023), while fake audio segments are generated with VITS (Kim et al., 2021) or YourTTS (Casanova et al., 2022). We select a subset of training and validation samples from the original training set (see App. A.1), and evaluate on 10k samples taken from the original validation set.

**DeepfakeEval 2024** (DFE-2024, Chandra et al. 2025) is a real-world dataset collected from 88 websites through social media and the `TrueMedia.org` platform. The dataset contains deepfakes circulating online in 2024 across multiple media (audio, video, images) and in 52 different languages; the types of manipulations are unknown. We use the video subset, comprising 45 hours of content, which we preprocess into single-speaker segments (see App. A.2). This process yields for the test set 86 real and 507 fake samples, averaging 13.92 seconds.

**AVLips** (AVLips, Liu et al. 2024b) is composed of 7557 videos (3373 real samples, 4184 fake samples). The real videos are obtained from LRS3 (Afouras et al., 2018), FF++ (Rossler et al., 2019) and DFDC (Dolhansky et al., 2020), while fake videos are generated using multiple methods

| Model | ID (AV1M) | | OOD | | | |
| | classif. | local. | FAVC | DFE-2024 | AVLips | mean |
|---|---|---|---|---|---|---|
| *Audio features* | | | | | | |
| AV-HuBERT (A) | 99.95 | 92.00 | **99.95** | 45.68 | **57.18** | 67.60 |
| Auto-AVSR (ASR) | 50.14 | 51.10 | 76.01 | 50.65 | 49.62 | 58.76 |
| Wav2Vec2 XLS-R 2B | **99.97** | **96.81** | 99.87 | **58.17** | 56.28 | **71.44** |
| *Visual features* | | | | | | |
| AV-HuBERT (V) | 93.67 | **94.53** | 95.53 | **60.74** | 90.45 | **82.24** |
| Auto-AVSR (VSR) | 58.98 | 65.14 | 77.53 | 59.48 | 70.13 | 69.08 |
| FSFM | 95.25 | 52.68 | 40.88 | 47.96 | 36.75 | 41.86 |
| CLIP VIT-L/14 | 96.54 | 80.17 | 95.18 | 48.66 | 53.26 | 65.70 |
| Video-MAE-large | **99.80** | 78.54 | 70.37 | 43.75 | 47.15 | 53.75 |
| *Audio-visual features* | | | | | | |
| AV-HuBERT (random init.) | 51.32 | 51.45 | 40.13 | 44.28 | 45.37 | 43.26 |
| AV-HuBERT | **99.88** | **96.80** | **99.52** | 50.87 | **84.35** | **78.24** |
| Auto-AVSR | 91.58 | 90.20 | 68.28 | **52.41** | 54.64 | 58.44 |

Table 2: Performance in terms of AUC of linear probes trained on AV1M and evaluated both in-domain (ID) and out-of-domain (OOD). For the in-domain setting, apart from classification we also consider the localization of the temporal explanations. Grayed-out values correspond to audio models evaluated on datasets with video-only manipulations, where labels may be unreliable.

such as MakeItTalk (Zhou et al., 2020), Wav2Lip (Prajwal et al., 2020), TalkLip (Wang et al., 2023) and SadTalker (Zhang et al., 2023). We use the whole dataset for testing.

## 5.2 EVALUATION AND BASELINES

**Classification.** To measure how well fake samples can be distinguished from real ones, we report the area under receiver operator characteristic curve (AUC). This metric has the advantages of being threshold-independent and having a clear baseline: a random model achieves 50% AUC, regardless of the class distribution. We also report average precision results in App. A.3, Tabs. A.3 & 5.

**Temporal localization.** We evaluate how well temporal explanations align with ground-truth annotations. For this, we consider the videos from the AV1M test set that contain at least one fake segment, and compute a localization score per video. The localization score is computed in terms of AUC by treating each frame as an independent sample: the model's frame-level score serves as the prediction, while annotated fake segment specifies the groundtruth label. The final localization score is the average of the AUC values across all fake test videos.

**Random baseline.** To understand how much information is encoded implicitly in the architecture, we report results of the the untrained (randomly initialized) AV-HuBERT model.

## 6 EXPERIMENTAL RESULTS

### 6.1 MAIN RESULTS

Tab. 2 shows the main results for the linear probes across the selected self-supervised features. We see that most representations perform strongly in-domain, with performances of over 90% AUC (the "classif." column). This result suggests that these features are able to encode the subtle information needed to differentiate between real and fake samples, and that these differences can be picked from different angles: from audio (Wav2Vec2, AV-HuBERT (A)), from motion information (Video-MAE), from static vision content (CLIP). The architecture does not have an implicit bias, as the randomly initialized AV-HuBERT model performs at random chance. There are only two other representations that do not perform well: the ASR and VSR models from Auto-AVSR.

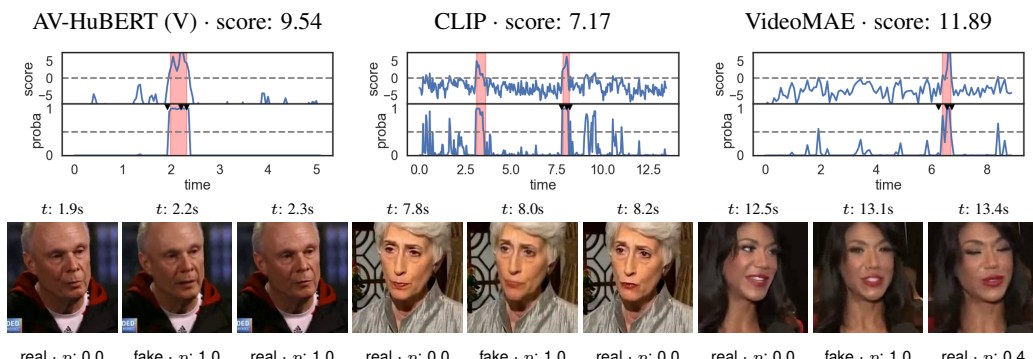

Figure 1: Temporal explanations of the top video predictions for three of the visual SSL representations: AV-HuBERT (V), CLIP, VideoMAE. The fakeness predictions are given in terms of unnormalized scores (logits), as well as as in terms of probabilities. Red regions indicate fake segments, and gray dashed lines corresponds to the decision boundary (0.5 probability). For each case, we show three frames: the central one taken from the fake region, and two that fall just outside the fake region (the precise locations are shown as triangles on the prediction plot).

Many of the features perform well in-domain, but is this information transferable? Or do the models overfit to some spurious biases? We tackle this question from three different perspectives: localization, out-of-domain performance, proxy tasks.

**Localization of temporal explanations.** Tab. 2 (column "local.") reports the alignment between the temporal explanations and local manipulations from AV1M. We see that many features still produce strong results (performance over 80% AUC), indicating that linear probes extract reasonable information from the representations. An exception is FSFM, for which there is a big drop, but visual inspection of the results didn't reveal an obvious bias that was picked up. Three other features are investigated qualitatively in Fig. 1. The explanations for the top scoring videos appear reasonable for these models, but the predictions of AV-HuBERT (V) are much cleaner than those of CLIP and VideoMAE. Individual samples reveal that the models attend to different cues: for example, CLIP tends to focus on blurry regions, while VideoMAE emphasizes speed-up or slow-down differences. However, these observations are difficult to quantify, since the perceptual differences are subtle and hard to gauge. Even between adjacent frames, which look very similar, the score can vary widely.

**Out-of-domain performance.** Tab. 2 (columns "OOD") reports results on three out-of-domain (OOD) datasets. Both ranking of features and the gap to the in-domain performance vary strongly across OOD datasets. In terms of feature ranking: on FAVC, which contains both audio and visual manipulations, audio features (Wav2Vec2, AV-HuBERT (A)) and audio-aligned features (AV-HuBERT) perform well; on DFE-2024 and AVLips, which primarily involve visual manipulations, the visual models perform best, with lip-focused models (AV-HuBERT (V), Auto-AVSR (VSR)) achieving the highest scores. In terms of generalization gap: on FAVC, some of the models produce performance that is comparable to what is reported on the in-domain case. However, the gap widens on the AVLips and DFE-2024 datasets. So while the models (like CLIP or AV-HuBERT) seem to learn reasonable features, they are not transferable to completely new datasets, likely because they contain different types of artifacts than those from AV1M or because they are harder (App. A.3, Tab. 6). Similar patterns are observed when FAVC is the source dataset (App. A.3, Tab. 7).

**Proxy tasks.** Tab. 3 presents the results on deepfake detection when models are trained on proxy tasks. Since the synchronization task involves two types of features, we also include those feature combinations for the supervised case (through late fusion) and for next-token prediction (through concatenation). Overall, proxy-task training performs worse than direct supervised training. This indicates that (i) the supervised classifier is not simply exploiting a spurious cue, and (ii) the proxy objectives themselves are less effective for the downstream task. Even on the harder DFE-2024 dataset, proxy tasks fail to provide an improvement (results not shown in the table). In terms of features, the AV-HuBERT (A) + (V) combination is the best, and the only one that is close to the supervised setting. For next-token prediction, AV-HuBERT (A) features are very strong, while for synchronization CLIP features are not enough, highlighting the importance of temporal dynamics.

| | AV1M | | | FAVC | | |
|---|---|---|---|---|---|---|
| Model | Sup. | NTP | Sync. | Sup. | NTP | Sync. |
| *Single features* | | | | | | |
| AV-HuBERT (A) | 99.95 | **90.57** | N/A | 99.95 | **80.54** | N/A |
| Wav2Vec2 | 99.97 | 56.62 | N/A | 99.87 | 59.36 | N/A |
| AV-HuBERT (V) | 93.67 | 46.14 | N/A | 95.53 | 55.32 | N/A |
| CLIP | 96.54 | 47.30 | N/A | 95.18 | 60.18 | N/A |
| *Combinations of features* | | | | | | |
| AV-HuBERT (A) + AV-HuBERT (V) | 99.97 | 84.47 | **87.28** | 99.95 | **91.23** | **96.11** |
| AV-HuBERT (A) + CLIP | 99.99 | **86.90** | 49.99 | 99.97 | 79.64 | 54.40 |
| Wav2Vec2 + AV-HuBERT (V) | 99.98 | 60.61 | 86.51 | 99.97 | 79.39 | 94.60 |
| Wav2Vec2 + CLIP | 99.99 | 57.17 | 49.66 | 99.96 | 69.75 | 21.67 |

Table 3: AUC performance on the AV1M and FAVC datasets for models trained with supervision (sup.), next-token prediction (NTP), and audio-video synchronization (sync.). The supervised models are trained on AV1M, so AV1M/Sup. results are in-domain (grayed out). The proxy models (NTP and sync.) are trained on a subset of VoxCeleb.

## 6.2 Combinations of features

To understand how the different self-supervised representations relate, we examine two aspects. First, we measure the correlation between predictions produced by pairs of models. This evaluation provides insight into the similarity of their learned decision boundaries. Second, we evaluate downstream performance when combining multiple models. This offers a more direct assessment of the feature combination effectiveness toward our final goal. Results on FAVC are in Fig. 2 and on DFE-2024 in App. A.3, Fig. 4.

**Model correlation.** We measure the correlation between their predictions. We consider a subset of the linear models trained on various features from the previous section. For each pair of models, we generate predictions on a shared test dataset and calculate the Pearson correlation coefficient between their outputs.

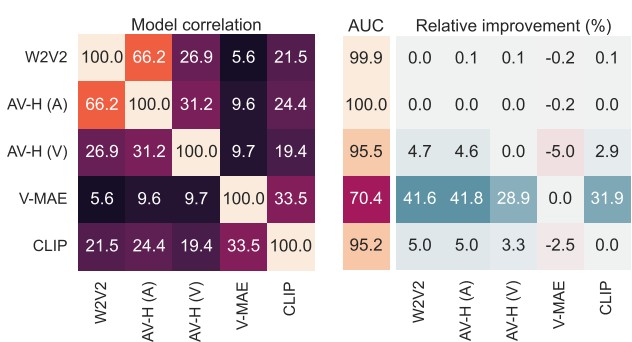

Figure 2: Correlations between models (left) and downstream performance (right). The downstream performance is presented in absolute values for the unimodal models (AUC column) and as relative improvement for feature combinations. Training was done on AV1M, testing on FAVC.

We observe that the cross-model correlations are generally weak to moderate; this suggests that the embeddings encode different information. The strongest correlations occur within modalities: audio models (AV-HuBERT (A) and Wav2Vec2) show high correlation with each other, as do vision models (CLIP and Video-MAE). Notably, AV-HuBERT (V) correlates more strongly with audio models than with other video models. This happens because AV-HuBERT (V) focuses solely on lip movements and is trained jointly with the audio component.

**Downstream performance.** The largest gains from feature combination occur for Video-MAE. This is expected, as Video-MAE has the lowest performance and thus has the most room for improvement. Also, unsurprisingly, Video-MAE benefits most from audio features, which are both stronger and more complementary than the visual ones. However, among similarly performing features, Video-MAE benefits more from CLIP (with which is more aligned) than from AV-HuBERT (V) (with which is more complementary). This suggests that the impact of feature combination is more nuanced than solely their complementarity.

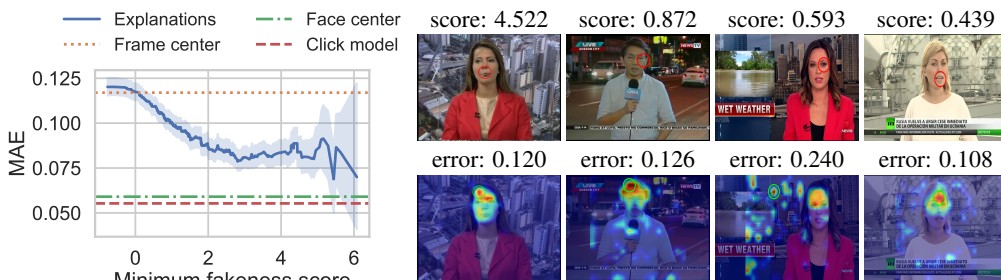

Figure 3: Alignment of spatial explanations to human annotations. *Left:* Alignment error in terms of mean absolute error (MAE) as a function of the model confidence (fakeness score). The explanations align better to human annotations as the model is more confident in its predictions. *Right:* Qualitative samples human annotation shown as the center of the red circle on top frame, and explanation of the CLIP-based model shown on bottom frame (maximum value indicated by the green circle).

## 6.3 ALIGNMENT TO HUMAN ANNOTATIONS

We saw that deepfake detection models tend to find temporarily modified regions. But do they also look at the same artifacts as humans? To answer this question, we analyze the recently introduced ExDDV dataset (Hondru et al., 2025), which provides click annotations indicating where humans identified generation artifacts. We compare these annotations with explanations produced by a CLIP-based model trained on the ExDDV training set (this model achieves 71.3% AUC on the ExDDV test set). We generate explanations for the fake test videos using GradCAM applied to the final LayerNorm layer of the CLIP visual encoder. We quantify the human-machine alignment as the mean absolute error (MAE) between the relative coordinates of human click annotations and the maximum values in the GradCAM attention maps. To contextualize these results, we compare against several baselines: random position within each frame, frame center, face center, and a predictive click model trained on click annotations (the ViT model from Hondru et al. 2025).

Fig. 3 (left) presents quantitative results: MAE as a function of the minimum fakeness score. We observe that error decreases with the minimum fakeness score, suggesting better alignment for confident predictions (these predictions are also correct, since we consider only fake samples). The variance increases correspondingly due to fewer samples exceeding the higher score thresholds. Compared to baseline methods, the explanations achieve better alignment than frame center (0.117 MAE) or random locations (0.270 MAE, not shown). However, alignment remains lower than what a predictive model achieves (0.055 MAE). Notably, the predictive click model performs only marginally better than face center prediction (0.058 MAE), suggesting that human annotations may not contain substantially more localization information beyond indicating that artifacts occur somewhere on the face. Fig. 3 (right) shows qualitative results. In these samples, most model predictions concentrate on the forehead region, while human annotations focus on eyes and lips. Crucially, the model appears to avoid relying on spurious background features, with explanations consistently concentrating on facial regions.

## 7 CONCLUSIONS

In this paper, we examined a wide array of self-supervised representations for audio-visual deepfake detection. We found that many of these features, varying by modality or training focus, perform strongly in-domain, with performance often aligning with temporal explanations. Moreover, the information captured by the representations is both complementary and often semantically meaningful. Among the tested representations, we have found that audio features (in particular Wav2Vec2) perform strongly when datasets contained speech-level manipulations, while AV-HuBERT provided the strongest overall performance. Nonetheless, even the best features degraded significantly as data distribution shifts, likely due to the presence of new types of artifacts. Taken together, our findings suggest that self-supervised features hold considerable promise for deepfake detection, but closing the generalization gap will likely require advances beyond feature representations alone.

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

# A APPENDIX

## A.1 IMPLEMENTATION DETAILS

**Linear probing.** The linear probing model is trained for 100 epochs with early stopping (the training process is stopped if no loss improvement on the validation set is observed for 10 consecutive epochs). The optimizer used is Adam with a learning rate of $10^{-3}$. In terms of data, for AV1M we select a subset of 45k videos as our training set and a subset of 5k videos for validation. Both subsets are sampled from the official training set. In the case of FAVC, we split the entire dataset into 63% training, 7% validation, and 30% testing samples.

**Next-token prediction.** We use similar training settings to linear probing. The main difference is that we anneal the learning rate using a cosine scheduler. To train the model we sampled 50k real videos from the training set of AV1M (45k for training, which includes the real ones from the train set of the linear probing experiments, and 5k for validation).

**Audio-video synchronization.** For the task of audio-video synchronization, we used the code made available from Smeu et al. (2025) with the default parameters. We utilized a temporal neighborhood of 30 frames and a learning rate scheduler with a patience of 5 epochs and a factor of 0.1, with a starting learning rate of $10^{-5}$. We used the same training set as for next-token prediction.

## A.2 DATASETS

**DeepfakeEval 2024 preprocessing.** We use TalkNet-ASD Tao et al. 2021 to identify which segments in an audio-video media file contain a single person speaking. We selected audio-video segments that met these criteria: (i) each video segment that has an associated audio stream; (ii) a video segment should contain a single speaking face that was tracked in every frame. (Some videos had static images or background music instead of speech and these were discarded); (iii) the identified face is larger than 100px×100px; (iv) the duration of audio-video segment is between 3 and 60 seconds.

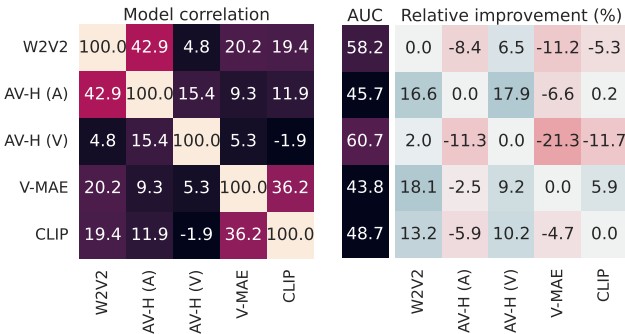

Figure 4: Correlations between models (left) and downstream performance (right). Training was done on AV1M, testing on DFEval-2024.

## A.3 FURTHER RESULTS

**Average precision results.** We complement the results from Tab. 2 and 3 with the average precision (AP) scores reported in Tab. A.3 and Tab. 5, respectively.

**Combinations of features.** In Fig. 4 we show the results for combinations of features when testing is done on DFE-2024. Compared to Fig. 2 (testing done on FAVC), certain trends are much better highlighted: first, the performance of every model greatly increases when combined with Wav2Vec2 or AV-HuBERT (V); second, the correlations between models' results are weaker, suggesting that the models capture more distinct patterns.

**In-domain performance.** In addition to the in-domain results on AV1M, we provide similar in-domain results on FAVC and DFE-2024. For DFE-2024, we use a training set composed of 532 videos (58 reals and 474 fakes) and a validation set of 134 samples (15 reals and 119 fakes). The average duration is 14.24 seconds. The performance obtained on the corresponding test splits is reported in Tab. 6. On FAVC, we observe high performance throughout (over 97% AUC), whereas on DFE-2024, performance hovers around 60% AUC across features, with FSFM features reaching the highest score of 69.68% AUC. These results indicate that DFE-2024 poses a considerably greater challenge even under in-domain evaluation.

**Out-of-domain performance.** We perform out-of-domain evaluation using a different source dataset: FAVC. The results are reported in Tab. 7. In comparison to Tab. 2, we see a greater generalization towards AVLips, especially in the visual-only features. Overall, the best features remain those extracted with AV-HuBERT.

| | ID (AV1M) | OOD | | | |
|---|---|---|---|---|---|
| Model | classif. | FAVC | DFE-2024 | AVLips | mean OOD |
| *Audio features* | | | | | |
| AV-HuBERT (A) | 99.95 | **99.99** | 85.25 | 62.88 | 82.70 |
| Auto-AVSR (ASR) | 51.47 | 98.43 | 87.53 | 55.30 | 80.42 |
| Wav2Vec2 XLS-R 2B | **99.97** | **99.99** | **89.91** | **64.32** | **84.74** |
| *Visual features* | | | | | |
| AV-HuBERT (V) | 94.68 | 99.58 | 88.40 | **88.59** | **92.19** |
| Auto-AVSR (VSR) | 60.25 | 98.68 | **89.48** | 74.33 | 87.49 |
| FSFM | 95.53 | 95.15 | 83.29 | 47.14 | 75.19 |
| CLIP VIT-L/14 | 96.78 | **99.76** | 84.17 | 60.86 | 81.59 |
| Video-MAE-large | **99.73** | 98.09 | 83.08 | 53.17 | 78.11 |
| *Audio-visual features* | | | | | |
| AV-HuBERT (random init.) | 52.46 | 94.22 | 84.00 | 51.66 | 76.62 |
| AV-HuBERT | **99.88** | **99.97** | 84.94 | **82.10** | **89.00** |
| Auto-AVSR | 92.79 | 97.67 | **87.31** | 58.39 | 81.12 |

Table 4: Performance in terms of AP of linear probes trained on AV1M and evaluated both in-domain (ID) and out-of-domain (OOD). We opt to omit the localization AP as this metric is sensitive to class imbalance, which makes it unreliable. Grayed-out values correspond to audio models evaluated on datasets with video-only manipulations, where labels may be unreliable.

| | AV1M | | | FAVC | | |
|---|---|---|---|---|---|---|
| Model | sup. | ntp | sync | sup. | ntp | sync |
| *Single features* | | | | | | |
| AV-HuBERT (A) | 99.95 | **91.81** | N/A | 99.99 | **98.76** | N/A |
| Wav2Vec2 | 99.97 | 56.40 | N/A | 99.99 | 96.58 | N/A |
| AV-HuBERT (V) | 94.68 | 49.16 | N/A | 99.58 | 97.13 | N/A |
| CLIP | 96.78 | 49.68 | N/A | 99.76 | 96.77 | N/A |
| *Audio-visual features* | | | | | | |
| AV-HuBERT (A) + AV-HuBERT (V) | 99.97 | 83.42 | **85.74** | 99.99 | **99.48** | **99.67** |
| AV-HuBERT (A) + CLIP | 99.99 | **87.98** | 50.02 | 99.99 | 98.55 | 96.14 |
| Wav2Vec2 + AV-HuBERT (V) | 99.98 | 59.43 | 83.31 | 99.99 | 98.57 | 99.61 |
| Wav2Vec2 + CLIP | 99.99 | 56.58 | 50.54 | 99.99 | 97.64 | 90.73 |

Table 5: AP performance on the AV1M and FAVC datasets for models trained with supervision (sup.), next-token prediction (NTP), and audio-video synchronization (sync.). The supervised models are trained on AV1M, so AV1M/Sup. results are in-domain (greyed out). The proxy models (NTP and sync.) are trained on a subset of VoxCeleb.

| | AV1M | | FAVC | | DFE-2024 | |
|---|---|---|---|---|---|---|
| Model | AUC | AP | AUC | AP | AUC | AP |
| *Audio features* | | | | | | |
| AV-HuBERT (A) | 99.95 | 99.95 | **100** | **100** | **65.53** | 90.78 |
| Auto-AVSR | 50.14 | 51.47 | 99.66 | 99.98 | 63.96 | **91.74** |
| Wav2Vec2 XLS-R 2B | **99.97** | **99.97** | **100** | **100** | 60.88 | 90.72 |
| *Visual features* | | | | | | |
| AV-HuBERT (V) | 93.67 | 94.68 | **100** | **100** | 66.32 | 92.08 |
| Auto-AVSR (VSR) | 58.98 | 60.25 | 97.83 | 99.89 | 61.43 | 90.12 |
| FSFM | 95.25 | 95.53 | 97.13 | 99.86 | **69.68** | **92.31** |
| CLIP VIT-L/14 | 96.54 | 96.78 | 99.77 | 99.98 | 64.37 | 89.54 |
| Video-MAE-large | **99.79** | **99.73** | 99.96 | 99.99 | 56.15 | 87.51 |

Table 6: Linear probes trained and evaluated in-domain (ID), reporting AUC and AP.

| Model | ID | OOD | | | |
| --- | --- | --- | --- | --- | --- |
| | FAVC | AV1M | DFE-2024 | AVLips | mean |
| *Audio features* | | | | | |
| AV-HuBERT (A) | **100** | **99.01** | 48.05 | 50.03 | 65.69 |
| Auto-AVSR (ASR) | 99.66 | 50.34 | 49.31 | **52.90** | 50.85 |
| Wav2Vec2 XLS-R 2B | **100** | 96.58 | **57.77** | 51.27 | **68.54** |
| *Visual features* | | | | | |
| AV-HuBERT (V) | **100** | 64.08 | **64.47** | **98.33** | **75.62** |
| Auto-AVSR (VSR) | 97.83 | 51.25 | 50.56 | 83.29 | 61.70 |
| FSFM | 97.13 | 52.69 | 64.36 | 84.29 | 67.11 |
| CLIP VIT-L/14 | 99.77 | **71.13** | 58.77 | 60.30 | 63.40 |
| Video-MAE-large | 99.96 | 60.01 | 44.19 | 71.31 | 58.50 |
| *Audio-visual features* | | | | | |
| AV-HuBERT | **100** | **94.50** | **57.41** | **78.47** | **76.79** |
| Auto-AVSR | 94.70 | 53.18 | 43.71 | 59.62 | 52.17 |

Table 7: Performance in terms of AUC of linear probes trained on FakeAVCeleb and evaluated both in-domain (ID) and out-of-domain (OOD). Grayed-out values correspond to audio models evaluated on datasets with video-only manipulations, where labels may be unreliable.