# OpenReview forum: "Investigating Self-Supervised Representations for Audio-Visual Deepfake Detection"
_ICLR.cc/2026/Conference — ICLR 2026 Conference Withdrawn Submission_

### Official Review · Reviewer_MybE · 2025-10-27

**Soundness:** 2
**Presentation:** 3
**Contribution:** 2
**Rating:** 4
**Confidence:** 4

**Summary:**

This paper evaluates self-supervised representations (SSL) for audio-visual deepfake detection through a systematic, cross-modal study. Using linear probing on frozen backbones from diverse modalities (audio, visual, and audiovisual), the authors analyze what information these representations encode, how interpretable their decisions are, and how complementary different modalities become when combined. Experiments across four datasets (AV-Deepfake1M, FakeAVCeleb, AVLips, and DeepfakeEval-2024) show that SSL features such as from CLIP, Wav2Vec2, and AV-HuBERT capture semantically meaningful deepfake cues and focus on manipulated facial regions rather than spurious artifacts. However, none of the models generalize reliably to unseen domains, showing that while SSL features can be strong and complementary, robust cross-dataset generalization remains a key open challenge.

**Strengths:**

The paper compares a sufficient spectrum of SSL models across modalities, providing a well-rounded analysis of their suitability for multimodal deepfake detection.

The linear-probe setup, log-sum-exp aggregation, and controlled training pipeline enable fair comparisons and isolate the representational power of the backbones.

The inclusion of temporal and spatial explanations, as well as quantitative comparison to human click annotations, goes beyond conventional performance reporting (e.g., simply AUC results) and provides some insight into model behavior.

The observation that SSL models encode meaningful yet non-transferable information, and that generalization failures stem largely from dataset characteristics rather than spurious feature reliance, can be useful for future work.

The writing is clear, and the paper is easy to understand.

**Weaknesses:**

The paper offers limited practical novelty in its insights. The main conclusion that self-supervised representations perform well in-distribution but fail to generalize is not very surprising, as this pattern is well established in the deepfake detection literature (see [1] for a popular earlier work), where most methods struggle under domain shift. It is thus largely expected that self-supervised models trained in a similar way would show the same behavior. While the analysis is systematic, it largely confirms known challenges rather than providing new conceptual understanding or solutions.

The related work section overlooks some prior studies exploring self-supervised learning for deepfake detection (e.g., [2], [3]). Moreover, Auto-AVSR is presented as a self-supervised method, whereas it is in fact a supervised audio-visual speech recognition model trained with labeled transcriptions. Additionally, earlier work such as LipForensics [4] already demonstrated that lipreading-based representations can be leveraged for deepfake detection.

The study relies on linear probes to evaluate representational quality for supervised learning. However, it is well known that Transformer-based SSL models often exhibit limited linear separability and can perform substantially better when fine-tuned in higher layers (e.g., [5]). This raises questions about how representative the reported results are of each model’s true potential, particularly with respect to generalization.

The paper does not compare its results to state-of-the-art deepfake detection methods. Without such baselines, it is difficult to interpret the absolute performance of the SSL features and to assess their practical relevance beyond relative comparisons.

[1] Li, Lingzhi, et al. "Face x-ray for more general face forgery detection." Proceedings of the IEEE/CVF conference on computer vision and pattern recognition. 2020.

[2] Haliassos, Alexandros, et al. "Leveraging real talking faces via self-supervision for robust forgery detection." Proceedings of the IEEE/CVF conference on computer vision and pattern recognition. 2022.

[3] Zhang, Daichi, et al. "Learning natural consistency representation for face forgery video detection." European Conference on Computer Vision. Cham: Springer Nature Switzerland, 2024.

[4] Haliassos, Alexandros, et al. "Lips don't lie: A generalisable and robust approach to face forgery detection." Proceedings of the IEEE/CVF conference on computer vision and pattern recognition. 2021.

[5] He, Kaiming, et al. "Masked autoencoders are scalable vision learners." Proceedings of the IEEE/CVF conference on computer vision and pattern recognition. 2022.

**Questions:**

Beyond confirming the well-known gap between in-distribution and out-of-distribution performance, what specific new insights do the authors believe this study provides about why self-supervised representations fail to generalize?

Since Transformer-based SSL models often exhibit weak linear separability but improve with fine-tuning, could the authors justify focusing just on linear probes (for supervised training)? Would partial fine-tuning change the generalization trends observed?

Would it be possible to include or discuss results against recent deepfake detectors, to contextualize the absolute performance of the evaluated SSL features?

---

### Official Review · Reviewer_ooUb · 2025-10-31

**Soundness:** 3
**Presentation:** 2
**Contribution:** 1
**Rating:** 2
**Confidence:** 3

**Summary:**

This paper investigates audio–visual deepfake detection using self-supervised representations. Unlike prior works that apply pretrained features within complex pipelines, the authors conduct a systematic evaluation across modalities and domains to analyze effectiveness, interpretability, and cross-modal complementarity.

**Strengths:**

- The paper is clearly written and easy to follow.

- The flow from problem definition to empirical observation is natural and coherent, allowing readers to easily grasp the motivation and key findings.

**Weaknesses:**

- The novelty and contribution are rather limited. The proposed methods are straightforward, and the paper focuses more on analysis than on developing new algorithms.

- While readability is good, the paper could be made more compact; for example, information such as Table 1 (basic model descriptions) belongs in the appendix rather than the main text.

- The comparisons are not fully appropriate for a deepfake detection study. The paper contrasts self-supervised models (e.g., AV-HuBERT vs. AV-HuBERT + proposed algorithm) rather than comparing against dedicated deepfake detection baselines.

Overall, while the motivation and storyline are strong, the technical content and experimental design do not yet meet the ICLR standard for methodological contribution.

**Questions:**

Please refer to Weakness.

---

### Official Review · Reviewer_ofoL · 2025-11-01

**Soundness:** 4
**Presentation:** 4
**Contribution:** 4
**Rating:** 8
**Confidence:** 4

**Summary:**

This paper investigates the use of self-supervised representations for audio-visual deep fake detection. Multiple audio and visual encoders are used and evaluated on multiple datasets and an attempt to identify the regions where these models focus on is made.

**Strengths:**

- The paper is well written
- The study presented is very interesting and novel (to the best of my knowledge)
- Results on multiple datasets are presented, both in-domain and out-of-domain
- The authors attempt to identify the regions the models attend to, which is very interesting.
- Overall, the conclusions are interesting.

**Weaknesses:**

- The auto-AVSR model performs better than AV-Hubert for visual and audio-visual speech recognition. Is there a hypothesis why it performs so much worse than AV-Hubert for deep fake detection.
- Table 2, the performance of audio and visual AutoAVSR features is much lower than the audio-visual AutoAVSR features. This looks a bit weird, since the gap for AV-Hubert is much smaller.
- Have the authors tried to combine features from multiple layers instead of using the last layer's features?
- One of the conclusions is that the proxy tasks (AV synchronisation and next-token prediction) do not work well for deep fake detection. Have the authors evaluate the performance of these models? i.e., how well they perform on AV synchronisation and next-token prediction. If they perform well, then the conclusion drawn is sound, if not, then no conclusion can be drawn.

**Questions:**

Please see above.

---

### Official Review · Reviewer_vUMH · 2025-11-03

**Soundness:** 2
**Presentation:** 3
**Contribution:** 1
**Rating:** 2
**Confidence:** 5

**Summary:**

The paper presents an empirical study of using pretrained self-supervised audiovisual representations as inputs to a simple deepfake detection head on top. The work uses standard audiovisual models trained on audiovisual speech datasets of human talking heads, and evaluates deepfake detection performance on standard downstream benchmarks.

While the paper is a detailed study of how these methods perform on the task, given the gaps in novelty and soundness (detailed below), I am leaning towards rejecting the paper.

**Strengths:**

- The chosen self-supervised audiovisual representation are appropriate, as are the evaluation datasets. The results presented on the tested datasets seem reasonable, and are accompanied by qualitative analysis of temporal and spatial explanations.

- The paper is well written and is easy to read and understand

- The reader could benefit from understanding how the chosen self-supervised method does on the standard datasets, which could help them not redo the same results in their own work.

**Weaknesses:**

- The paper uses linear probing on top of pretrained self-supervised audiovisual models to evaluate on multiple downstream datasets. To me, this is akin to running a bunch of baselines on the tested datasets with no particular technical novelty.

- A crucial technical drawback of the evaluation setting to me is the fact that individual frames are encoded separately by the audio/visual backbones before being aggregated. While frame-level information does provide useful information, the essence of audiovisual deepfake detection (for the more complex synthetic examples) is usually tied to the temporal patterns of the data (both synchrony between lip/face movements with the speech content and its consistency and realness over time). Instead of embeddings extracted from one frame alone, at the bare minimum I would have liked to see an ablation study using a sliding window or being modeled temporally using a token mixing Transformer or recurrent layer.

- Given that this work was more focused on evaluating existing representations, I found the lack of methods like AVData2Vec and RAVEN to be a gap. These methods are similar to AV-Hubert but presented better performance on related audiovisual understanding tasks.

- The paper does not effectively introduce seminal related work in important areas. For example, for audiovisual synchronization, works like SyncNet (Chung et al), Multisensory Networks (Owens et al) were important. Similarly, I found no introduction of work related to synthetic talking head generation / speech-driven facial animation (even though it is mentioned in the fake video generation portion of the chosen datasets).

- Line 1 of the abstract is “Self-supervised representations excel at many vision and speech tasks, but their potential for audio-visual deepfake detection remains underexplored” -> For many of the related papers that the work cites, the baselines seem to be self-supervised representations!

- I think the overall writing can be made clearer to indicate that the focus is on human-centric “talking head” videos as opposed to general deepfake videos. Given this, some of the qualitative analysis showing heightened contributions from the facial regions seems unnecessary to me (and the spurious detections in the non facial regions are potentially problematic).

**Questions:**

- Did the authors try to preprocess the datasets to only have the face crops used as input? If so, did they try to do any preprocessing (e.g. face alignment) that is commonly used in these pipelines?

Also, it would be helpful to directly address the weaknesses I mentioned above.

---

### Note · Authors · 2025-11-13

**Comment:**

We would like to withdraw this submission as we are preparing a significantly improved version.

**Withdrawal Confirmation:**

I have read and agree with the venue's withdrawal policy on behalf of myself and my co-authors.